# Association of Structural Social Capital and Self-Reported Well-Being among Japanese Community-Dwelling Adults: A Longitudinal Study

**DOI:** 10.3390/ijerph18168284

**Published:** 2021-08-05

**Authors:** Kazuya Nogi, Haruhiko Imamura, Keiko Asakura, Yuji Nishiwaki

**Affiliations:** 1Department of Environmental and Occupational Health, Toho University Graduate School of Medicine, Ota-ku, Tokyo 143-8540, Japan; md18021n@st.toho-u.jp; 2Department of Environmental and Occupational Health, School of Medicine, Toho University, Ota-ku, Tokyo 143-8540, Japan; keiko.asakura@med.toho-u.ac.jp (K.A.); yuuji.nishiwaki@med.toho-u.ac.jp (Y.N.)

**Keywords:** social capital, local community activities, neighborhood relationships, well-being, happiness, self-rated health, depressive symptoms

## Abstract

Previous studies have shown both positive and non-positive associations between social capital and health. However, longitudinal evidence examining its comprehensive effects on well-being is still limited. This study examined whether structural social capital in the local community was related to the later well-being of Japanese people aged 40 or above. A 4-year longitudinal study was conducted in a rural Japanese town. “Well-being” was measured using three indicators (happiness, self-rated health, and depressive symptoms), and those who were high in well-being in the baseline 2015 survey and responded to the follow-up 2019 survey were analyzed (*n* = 1032 for happiness, 938 for self-rated health, and 471 for depressive symptoms). Multilevel Poisson regression analysis adjusted for covariates showed that having contact with fewer neighbors was associated with a decline in happiness at both the community level (adjusted relative risk = 1.64, 95% confidence interval = 1.20–1.63) and the individual level (adjusted relative risk = 1.51, 95% confidence interval = 1.05–2.17), but participation in local community activities was not. The results suggest that dense personal networks might be more important in areas with thriving local community activities, not only for individuals but also for all community members.

## 1. Introduction

In the Preamble to the Constitution of the World Health Organization, health is defined as “a state of complete physical, mental, and social well-being and not merely the absence of disease or infirmity” [1]. To achieve health, doctors have traditionally emphasized treating physical diseases [2]. However, the fact that physically healthy people are not always happy is a common enough observation. It is essential to seek mental and social well-being too, not only because health consists of physical, mental, and social well-being, but also because these factors influence each other. For example, subjective well-being, which is an individual’s positive feelings or evaluation of their own lives, is known as a protective factor for physical health. Individuals with high subjective well-being tend to have a lower incidence rate of cardiovascular diseases, lower mortality, better immune function, and faster wound healing [3,4,5]. Gaining good mental and social well-being is another way to achieve health, apart from treating physical disease.

Recently, the association between social capital and health has become a major area of interest in the field of social medicine [6]. Social capital is referred to as the resource that individuals can gain from the social connections surrounding them. Social capital comprises both structural and cognitive components. Structural social capital refers to actual participation in social networks, such as neighborhood associations and volunteer groups, whereas cognitive social capital refers to perceptions of belonging to networks and has often been measured by trust and reciprocity [6,7,8]. There are three types of social capital depending on the nature of the network: bonding, bridging, and linking social capital. Bonding social capital refers to the resources within groups or networks of members with similar characteristics. In contrast, bridging social capital refers to resources from the connection between different kinds of groups or individuals. Linking social capital can be gained from access to formal or institutionalized power or authority structures [6,8,9]. Among the multiple dimensions of social capital, it might be more practical to intervene in structural social capital since it is easier to provide opportunities to participate in such networks than to change people’s perceptions about themselves. Additionally, if the norms of reciprocity and trustworthiness arise from social networks as Putnam mentioned [10] (p. 19), structural social capital might be located upstream of social capital.

Previous studies have shown a positive association between various aspects of social capital and health [11]. For example, using cross-national data from the World Values Survey and the large-scale surveys in the US and Canada, Helliwell and Putnam found that civic engagement, trust, and social ties were positively associated with subjective well-being [12]. In a cross-sectional survey targeting 8028 middle-aged and older adults, Nieminen et al. found that social participation was positively associated with self-rated health (SRH) [13]. However, researchers have considered the possibility of negative associations, particularly in bonding social capital [6,8,14,15]. For example, Mitchell and LaGory reported that higher bonding social capital at the individual level was associated with mental distress in an impoverished community in a southeast US city [16], while Kim et al. showed that bonding social capital was associated with better SRH at both the community and individual levels in 40 US communities [17]. Villalonga–Olives and Kawachi reviewed 44 articles that reported the non-positive effects of social capital on health and indicated that “several downsides of social capital seem to occur in the context of strong bonding capital, but weak bridging capital” [18] (p. 126). The impact of social capital on health might be inconsistent depending on the community or group characteristics and aspects of the social capital; however, few studies have simultaneously examined its features in community life. Additionally, most previous studies have been conducted in western countries, and longitudinal evidence in Japan that examines comprehensive effects on well-being is limited [19].

In this study, we focused on structural social capital in the local community: participation in local community activities and neighborhood relationships. As mentioned above, it could be easier to intervene in structural social capital than cognitive social capital. Although our society is increasingly connected by sophisticated transportation and high-speed information networks, community-based relationships remain important, especially in local (non-urban) areas where people do not frequently move in or out. In such areas, there are longstanding local community activities that serve as formal and bonding social capital in many cases. Furthermore, there are close neighborhood relationships, which are rather less formal and sometimes bridging social capital. We hypothesized that people who participate in local community activities and have many neighbors would have higher well-being. The objective of this study was to examine the relationship between social capital (participation in local community activities and neighborhood relationships) and people’s self-reported well-being using longitudinal data. This could help policymakers and community leaders to choose interventions to improve the well-being of residents more effectively.

## 2. Materials and Methods

### 2.1. Study Population

This longitudinal study was part of a questionnaire survey on health and daily life conducted in Koumi Town, Nagano Prefecture, Japan. It is a relatively small rural town located in a mountainous area with a population of approximately 4500. This questionnaire survey was conducted as one of the town’s projects for the prevention of lifestyle-related diseases, and it targeted all residents aged 40 or above who were not certified for nursing care. The baseline survey conducted in January 2015 was followed up in January 2019. In 2015, we distributed the questionnaire to 3112 residents with the support of Koumi Town Hall and health promotion volunteers, and received 2105 responses (a response rate of 67.6%) (Figure 1).

In this study, we set three well-being-related indicators: happiness, SRH, and depressive symptoms. Happiness refers to one’s positive feelings or evaluation of his/her life and was measured by the following question: “Do you feel you are happy?” It approximates the Japanese General Social Surveys project question [20] and refers to “overall happiness” rather than specific components of subjective well-being such as “life satisfaction” or “positive/negative affect” [21]. While multi-item measures are necessary to analyze the components of subjective well-being, such single-item measures are the method of choice when dealing with “overall happiness” [22]. Response possibilities were “happy,” “rather happy,” “neither,” “not so happy,” and “not happy at all.” We considered “happy” and “rather happy” as “high” well-being, and the other three responses as “low,” based on a previous study [21]. Additionally, SRH is a subjective assessment that focuses more on the physical aspect. It was measured by the question: “How is your current health status?” This type of single-item measure is a well-established way to capture one’s self-reported general health and has been used in many surveys [23]. Response possibilities were “excellent,” “good,” “fair,” “poor,” and “very poor.” We considered the first two responses as “high” well-being and the other three as “low,” based on a previous study [24]. Depressive symptoms focused on feelings that showed signs of depression. These were assessed using the Kihon Checklist, which was developed by Japan’s Ministry of Health, Labour, and Welfare to screen elderly individuals who are at high risk of disability [25]. It includes five items to reflect depressive symptoms: “(In the last two weeks) have you felt a lack of fulfillment in your daily life?”, “(In the last two weeks) have you felt a lack of joy when doing the things you used to enjoy?”, “(In the last two weeks) have you felt difficulty in doing what you could do easily before?”, “(In the last two weeks) have you felt helpless?”, and “(In the last two weeks) have you felt tired without a reason?” The criterion for a high risk of depression in the Kihon Checklist is when two or more of these items are met. We considered those at low risk for depression to have “high” well-being and those at high risk to have “low” well-being as assessed by the Kihon Checklist. Since the Kihon Checklist was intended for individuals aged 65 or above, depressive symptoms were assessed for these older adults only.

To analyze self-reported well-being, we excluded the responses of someone other than the participant him/herself (*n* = 228) from main analyses using the answer to the question about who filled out the questionnaire. Those who indicated “high” well-being at the baseline survey were eligible for the study (*n* = 1470 for happiness, 1300 for SRH, and 687 for depressive symptoms). Of those, respondents who answered indicators about well-being in the follow-up 2019 survey were analyzed; the study population included 1032 for happiness, 938 for SRH, and 471 for depressive symptoms (Figure 1).

### 2.2. Outcome Measurement

The outcome was a decline in well-being from “high” in the baseline 2015 survey to “low” in the follow-up 2019 survey for each related indicator: happiness, SRH, and depressive symptoms. Measurements of these indicators and definitions of “high”/“low” well-being have been described above.

### 2.3. Social Capital Measurement

In this study, we measured two aspects of structural social capital in the local community in the baseline 2015 survey: participation in local community activities and the number of neighbors in contact. These two items were assessed in a survey conducted by the Cabinet Office of the government of Japan in 2003 [26], and the questions in our survey followed those. Each participant was assessed at the individual and community levels. There were 34 town areas in Koumi at the time of the baseline survey. We aggregated the individual responses of social capital (study population for the community-level social capital; *n* = 2105) to community-level social capital for each area.

Participation in local community activities was assessed by the frequency of attending events of any local group, such as neighborhood associations, children’s gatherings, and fire brigades. Seven possible responses in the questionaries were “do not participate,” “a few days a year,” “one day a month,” “two or three days a month,” “one day a week,” “two or three days a week,” and “not less than four days a week.” Participation in local community activities was classified into two categories: “participate” and “do not participate.” We defined “participate” as the frequency of at least “a few days a year” based on a previous study [27]. The prevalence of “participate” was calculated for each town area and dichotomized into “high” (top 17 areas) and “low” (bottom 17 areas) social capital at the community level.

For the number of neighbors in contact, the following five response possibilities were provided: “don’t know my neighbors,” “have contact with just a few neighbors (less than five),” “have contact with some neighbors (between five and nine),” “have contact with quite a lot of neighbors (between ten and 19),” and “have contact with a lot of neighbors (20 or more).” For the analysis, the number of neighbors in contact was classified into two categories: “small” and “large.” We defined “small” as less than five based on previous studies [26,28] In Koumi, the research population of each town area varied from 12 to 126 people. As a consequence of classification by the set criteria, residents could be classified into two groups at a reasonable proportion even in small population areas (Table A1). The prevalence of “large” was calculated for each town area and dichotomized into “high” (top 17 areas) and “low” (bottom 17 areas) social capital at the community level.

### 2.4. Covariates

Age, sex, marital status, cohabitants, educational attainment, history of major diseases, current drinking, and current smoking were obtained from the baseline 2015 survey as covariates. In the analyses for happiness and SRH, age was categorized into “40s,” “50s,” “60s,” “70s,” and “80 years or older.” In the analyses for depressive symptoms, age was a continuous variable, since only individuals aged 65 or above were analyzed. History of major diseases was defined as having any one of the following diseases known to cause death or disability: stroke, myocardial infarction/angina, diabetes, vertebral or femoral neck fracture, Parkinson’s disease, or cancer. Other covariates were divided into two categories each: sex (“female” vs. “male”), marital status (“married” vs. “not married”), cohabitants (“living with someone” vs. “living alone”), educational attainment (“≥10 years” vs. “<10 years”), history of major diseases (“no” vs. “yes”), current drinking (“no” vs. “yes”), and current smoking (“no” vs. “yes”).

### 2.5. Statistical Analysis

The relative risks (RRs) of a decline in well-being and related 95% confidence intervals (CIs) were estimated for each well-being-related indicator using multilevel modified Poisson regression analysis to assess whether community- and individual-level social capital (participation in local community activities and the number of neighbors in contact) at baseline were associated with the outcome. Since we used a dataset with a hierarchical structure (individuals were nested within the town areas), we performed a multilevel analysis, which provided robust results. The modified Poisson regression approach (Poisson regression with a robust error variance) was developed to rectify variance overestimation that occurs when Poisson regression is applied to binary data and can be used to estimate the relative risks [29]. Three analysis models were created. First, social capital at the community level (participation in local community activities and the number of neighbors in contact), age, and sex were included in the model (Model 1). Then, social capital at the individual level (participation in local community activities and the number of neighbors in contact) was added (Model 2). Finally, marital status, cohabitants, educational attainment, history of major diseases, current drinking, and current smoking were added (Model 3). Those who had missing values for covariates were excluded from each model. Additionally, the analysis stratified by sex was conducted for happiness and SRH (for depressive symptoms, the number was insufficient to perform subgroup analysis).

As for the sensitivity analysis, we conducted one in which the number of neighbors in contact was assessed using the original five items: “don’t know my neighbors,” “have contact with just a few neighbors (less than five),” “have contact with some neighbors (between five and nine),” “have contact with quite a lot of neighbors (between ten and 19),” and “have contact with a lot of neighbors (20 or more).”

Statistical significance was set at *p* < 0.05. All analyses were performed using STATA version 16.1 (STATA Corporation, College Station, TX, USA).

## 3. Results

The mean age (standard deviation) of the study population at the baseline 2015 survey was 65.4 (11.3) years in the analysis of happiness, 65.3 (11.2) years in the analysis of SRH, and 73.0 (6.0) years in the analysis of depressive symptoms. Table 1 shows the characteristics of the study population for each well-being-related indicator. Regarding social capital at the individual level, slightly more than half of the respondents participated in local community activities (54.3% in the analysis of happiness, 55.5% in the analysis of SRH, and 57.4% in the analysis of depressive symptoms), and many more had at least five neighbors in contact (86.6% in the analysis of happiness, 85.5% in the one of SRH, and 89.1% in the one of depressive symptoms).

Regarding the community-level social capital, the prevalence of “participate” in local community activities and “large” number of neighbors in contact ranged from 14.3% to 78.3% (the range of the top 17 areas was 39.6% and over), and from 58.8% to 89.9% (the range of the top 17 areas was 73.8% and over), respectively (Table A1).

Table 2 shows the association of social capital at the community and individual levels with a decline in well-being. During the follow-up 4 years, the incidence of the decline was 10.6% (109/1032) for happiness, 20.1% (189/938) for SRH, and 16.8% (79/471) for depressive symptoms. After adjusting all covariates (Model 3), the number of neighbors in contact was significantly associated with the decline in happiness at both the community and individual levels; adjusted RR of “low” social capital assessed by the number of neighbors in contact was 1.64 (95% CI: 1.20–1.63) at the community level and 1.51 (95% CI: 1.05–2.17) at the individual one. The association between participation in local community activities and a decline in happiness was not clear. Among the covariates, marital status and current drinking were associated with happiness; adjusted RR was 2.05 (95% CI: 1.31–3.20) and 0.60 (95% CI: 0.38–0.94), respectively (data not shown). Regarding SRH and depressive symptoms, the measured social capital did not show any association with the decline in well-being, at either the community or individual levels. In the sex-stratified analysis, there was a difference in the association between the number of neighbors in contact at the individual level and happiness. While the association was almost null in men (RR in Model 3: 1.06, 95% CI: 0.46–2.45), a clear risk was observed in women (RR in Model 3: 1.73, 95% CI: 1.07–2.79). However, there was no statistical interaction between sex and this association (*p* = 0.352). Regarding SRH, no sex differences were observed.

The sensitivity analysis showed a similar trend to the main results (Table 3); an association between the number of neighbors in contact at the individual level and the decline in happiness was observed when the number of neighbors was less than five.

## 4. Discussion

In this study, we observed a positive association between the number of neighbors in contact and happiness at both the community and individual levels. However, the association of participation in local community activities with happiness was not clear. SRH and depressive symptoms were not associated with these two aspects of social capital.

To the best of our knowledge, no previous longitudinal studies have focused on the association between the number of neighbors in contact and individual well-being. One cross-sectional study used a similar indicator of neighborships as ours and reported a positive association with happiness [28]. Bowling et al. found that good subjective assessment or feeling about the neighborhood environment was associated with good SRH and physical functioning [30]. Since those with more neighbors in contact are socialized people and likely to have a good impression about their community, it might be possible that the number of neighbors in contact acted as a proxy for perceptions of the neighborhood environment. This simple quantitative indicator might be suitable for research in analytic epidemiology, although further studies are needed to verify its external validity.

Regarding the association between local community activities and well-being, the results of a longitudinal study by Fancourt and Steptoe were similar to those of the current one [31]. They examined which type of group participation was associated with subjective well-being and found that participation in education/art/music classes and church/religious groups was positively associated with it, while participation in community-based activities such as resident groups or neighborhood watch groups was not associated. Portes identified four pathways through which social capital can have a negative impact: the exclusion of outsiders, excessive claims on group members, restrictions on individual freedoms, and downward leveling norms [15]. Koumi has thriving local community activities, and many of these have been held in each town area for a long time. We hypothesized that such traditional social ties would constitute an imposition on individuals. However, the neighborhood relationship is more informal and personal. Given that existing organizations are very formal, residents might experience more satisfaction from open relationships they themselves have formed. If so, spontaneous gatherings might enhance the free interaction between residents and improve their well-being. For example, by creating new free spaces or readily accessible cafes, different kinds of networks that do not rely on traditional community activities could be formed. Recently, policies to tackle social isolation have been initiated, such as the establishment of the Minister for Loneliness in the UK and Japan. The results of this study constitute a rationale for these policies.

In the current study, we measured three well-being-related indicators and found a clear association with social capital only for happiness. There are three possible reasons for this discrepancy. First, the effect of social capital on SRH might differ between the structural and cognitive varieties. A previous study of five-year longitudinal data examined social trust and civic participation at both the individual and community levels, and only social trust was found to have a significant association with SRH [32]. Another cross-sectional study from Japan focused on social capital at the community level and examined ten components of social capital, and found that SRH was not associated with structural social capital, but only with cognitive social capital [33]. Cognitive social capital might have more influence on SRH than structural social capital, but the current study could not assess this. Future studies should clarify the impact of both structural and cognitive social capital on SRH. Second, the follow-up period may have been too short. Eriksson and Ng reported that remaining at a low level of social participation or worsening over 10 years was associated with poor SRH [7]. The current study followed a relatively short period of 4 years. A longer follow-up period may be needed to detect the association between structural social capital and SRH. Third, for depressive symptoms, the number of subjects in the analysis might have been too small. A previous study examined data from 29,065 people over 65 and reported that civic participation at both the community and individual levels was inversely related to the onset of depressive symptoms [34]. In contrast, we recruited only 471 residents for the analysis of depressive symptoms because those under 65 were not covered by the relevant questions.

This study had three major strengths. First, we conducted a multilevel analysis using longitudinal data. To assess social capital, which is naturally influenced by where individuals live, multilevel analysis is a well-established method [35]. Additionally, the community- and individual-level indicators of social capital were included in the same model in the current study. It showed that having a large number of neighbors in contact at both the community and individual level had a positive impact on happiness. In other words, dense personal networks might be beneficial for all community members both individually and collectively. Second, the current study included middle-aged and elderly individuals (aged 40 or above). In Japan, the Japan Gerontological Evaluation Study group has conducted several longitudinal studies that examined the association between social capital and health; however, these only included older adults [34,36,37,38,39,40]. Third, we excluded proxy responses to evaluate pure self-assessments of well-being [41].

This study had several limitations. First, happiness was measured by the single-item scale so that specific aspects of subjective well-being such as “life satisfaction” or “positive/negative affect” could not be assessed [42]. In future studies, a multi-item scale should be used to capture various aspects of well-being. However, we reinforced the analysis by adding SRH and depressive symptoms as related but different indicators of well-being. Happiness focuses on the emotional aspect of self-assessment, while SRH on the physical aspect and depressive symptoms on the signs of the disease. The result that the benefit of social capital was shown only for happiness indicates that, at least in the short term, the impact on the emotional aspect is stronger. Second, we were not able to adjust for the economic status that was not included in the questionnaire items. Economic status was thought to be an important factor for individual well-being. Therefore, we adopted educational attainment, which is strongly related to economic status, as a covariate [43]. Third, the study population did not include those who answered in the 2015 baseline survey but not in the 2019 follow-up survey, which means that there could be a selection bias. Many of those who were not included in the analysis had a low number of neighbors in contact at the baseline survey. If we assume that they were unable to respond in the follow-up survey due to a decline in well-being, the effect of bias would be an underestimation of the main results and does not change the interpretation of the results. Fourth, the questionnaire survey was designed for those aged 40 or above and did not include younger people. The patterns of participation in local community activities and neighborhood relationships for younger people may differ from those for middle-aged and elderly people, and these should be examined in the future. Finally, this study was based on a questionnaire survey conducted in a single rural town in Japan. Further studies are needed to determine whether similar results can be obtained in urban areas.

## 5. Conclusions

Among the structural social capital in the local community, the number of neighbors in contact was positively associated with happiness at both the community and individual levels. The results suggest that dense personal networks might be more important in areas with thriving local community activities, not only for individuals but for all community members as well. Interventions that encourage individuals to connect with their neighbors outside of (historically) existing local community activities might improve their well-being.

## Figures and Tables

**Figure 1 ijerph-18-08284-f001:**
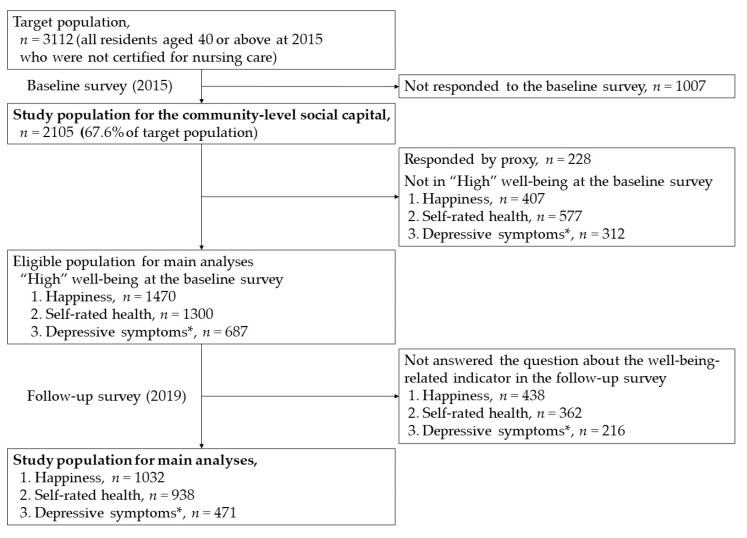
Study population. * Depressive symptoms were assessed for those aged 65 or above.

**Table 1 ijerph-18-08284-t001:** The baseline characteristics of the study population for each well-being-related indicator.

Characteristics			Well-Being-Related Indicator					
			Happiness		Self-Rated Health		Depressive Symptoms	
			*n* = 1032		*n* = 938		*n* = 471	
			*n* (%)		*n* (%)		*n* (%)	
**Social capital at the community level**								
	Participation in local community activities	High	544	(52.7)	486	(51.8)	237	(50.3)
		Low	488	(47.3)	452	(48.2)	234	(49.7)
	The number of neighbors in contact	High	554	(53.7)	503	(53.6)	275	(58.4)
		Low	478	(46.3)	435	(46.4)	196	(41.6)
**Social capital at the individual level**								
	Participation in local community activities	Participate	543	(54.3)	506	(55.5)	260	(57.4)
		Do not participate	457	(45.7)	405	(44.5)	193	(42.6)
	The number of neighbors in contact	Large (≥5)	881	(86.6)	790	(85.5)	415	(89.1)
		Small (<5)	136	(13.4)	134	(14.5)	51	(10.9)
**Covariates**								
	Age (years)	40–49	116	(11.4)	107	(11.5)	0	(0.0)
		50–59	184	(18.0)	159	(17.1)	0	(0.0)
		60–69	322	(31.5)	299	(32.2)	153	(32.5)
		70–79	289	(28.3)	270	(29.1)	245	(52.0)
		≥80	110	(10.8)	93	(10.0)	73	(15.5)
	Sex	Female	585	(56.9)	518	(55.5)	260	(55.2)
		Male	444	(43.1)	416	(44.5)	211	(44.8)
	Marital status	Married	861	(84.6)	761	(82.1)	383	(81.8)
		Not married	157	(15.4)	166	(17.9)	85	(18.2)
	Cohabitants	Yes	951	(92.8)	854	(91.8)	423	(89.8)
		No	74	(7.2)	76	(8.2)	48	(10.2)
	Educational attainment (years)	≥10	790	(77.1)	719	(77.2)	313	(66.7)
		<10	234	(22.9)	212	(22.8)	156	(33.3)
	History of major diseases *	No	861	(83.4)	795	(84.8)	378	(80.3)
		Yes	171	(16.6)	143	(15.2)	93	(19.7)
	Current drinking	No	496	(48.9)	451	(48.9)	267	(57.8)
		Yes	519	(51.1)	472	(51.1)	195	(42.2)
	Current smoking	No	886	(89.0)	805	(89.0)	423	(92.6)
		Yes	109	(11.0)	100	(11.0)	34	(7.4)

* History of major diseases was defined as having any one of the following diseases: stroke, myocardial infarction /angina, diabetes, vertebral or femoral neck fracture, Parkinson’s disease, or cancer.

**Table 2 ijerph-18-08284-t002:** The association of social capital at the community and individual levels with a decline in well-being: results from multilevel Poisson regression models.

			Outcome/Study Population (%)		Model 1 *			Model 2 †			Model 3 ‡		
					Adjusted RR (95% CI)		*p*-Value	Adjusted RR (95% CI)		*p*-Value	Adjusted RR (95% CI)		*p*-Value
**Social capital associated with a decline in happiness**													
Total analyzed population			109/1032	(10.6%)	*n* = 1019			*n* = 976			*n* = 929		
at the community level													
	Participation in local community activities	High	49/544	(9.0%)	1.00			1.00			1.00		
		Low	60/488	(12.3%)	1.23	(0.95–1.59)	0.117	1.18	(0.90–1.55)	0.220	1.21	(0.90–1.63)	0.209
	The number of neighbors in contact	High	43/554	(7.8%)	1.00			1.00			1.00		
		Low	66/478	(13.8%)	1.76	(1.32–2.34)	<0.001	1.58	(1.15–2.17)	0.005	1.64	(1.20–2.22)	0.002
at the individual level													
	Participation in local community activities	Participate	50/543	(9.2%)				1.00			1.00		
		Do not participate	52/457	(11.4%)				1.02	(0.72–1.46)	0.901	0.95	(0.67–1.36)	0.799
	The number of neighbors in contact	Large (≥ 5)	82/881	(9.3%)				1.00			1.00		
		Small (< 5)	23/136	(16.9%)				1.61	(1.10–2.37)	0.015	1.51	(1.05–2.17)	0.027
**Social capital associated with a decline in self-rating health**													
Total analyzed population			189/938	(20.1%)	*n* = 925			*n* = 888			*n* = 847		
at the community level													
	Participation in local community activities	High	92/486	(18.9%)	1.00			1.00			1.00		
		Low	97/452	(21.5%)	1.16	(0.87–1.54)	0.320	1.15	(0.86–1.54)	0.332	1.17	(0.86–1.58)	0.320
	The number of neighbors in contact	High	98/503	(19.5%)	1.00			1.00			1.00		
		Low	91/435	(20.9%)	1.08	(0.81–1.44)	0.593	1.02	(0.74–1.40)	0.896	1.00	(0.72–1.39)	0.982
at the individual level													
	Participation in local community activities	Participate	106/506	(20.9%)				1.00			1.00		
		Do not participate	75/405	(18.5%)				0.84	(0.62–1.13)	0.254	0.82	(0.61–1.12)	0.210
	The number of neighbors in contact	Large (≥ 5)	152/790	(19.2%)				1.00			1.00		
		Small (< 5)	33/134	(24.6%)				1.30	(0.87–1.94)	0.197	1.28	(0.84–1.96)	0.249
**Social capital associated with an increased risk of** **depressive symptoms**													
Total analyzed population			79/471	(16.8%)	*n* = 471			*n* = 448			*n* = 432		
at the community level													
	Participation in local community activities	High	42/237	(17.7%)	1.00			1.00			1.00		
		Low	37/234	(15.8%)	0.91	(0.59–1.40)	0.661	0.82	(0.55–1.23)	0.345	0.79	(0.51–1.22)	0.292
	The number of neighbors in contact	High	47/275	(17.1%)	1.00			1.00			1.00		
		Low	32/196	(16.3%)	0.97	(0.65–1.44)	0.875	0.86	(0.58–1.27)	0.440	0.77	(0.51–1.15)	0.199
at the individual level													
	Participation in local community activities	Participate	36/260	(13.8%)				1.00			1.00		
		Do not participate	38/193	(19.7%)				1.51	(0.96–2.39)	0.077	1.46	(0.88–2.41)	0.144
	The number of neighbors in contact	Large (≥5)	68/415	(16.4%)				1.00			1.00		
		Small (<5)	10/51	(19.6%)				1.22	(0.63–2.35)	0.549	1.13	(0.58–2.18)	0.718

Note: RR = relative risk; CI = confidence interval; * Model 1. Social capital at the community level adjusted for age and sex; † Model 2. Social capital at the community and individual levels adjusted for age and sex; ‡ Model 3. Social capital at the community and individual levels adjusted for age, sex, marital status, cohabitants, educational attainment, history of major diseases (having any one of the following diseases: stroke, myocardial infarction /angina, diabetes, vertebral or femoral neck fracture, Parkinson’s disease, or cancer), current drinking and current smoking.

**Table 3 ijerph-18-08284-t003:** The association of the number of neighbors in contact at the individual level with the decline in well-being.

The Number of Neighbors in Contact at the Individual Level	Well-Being Related Indicator											
	Happiness				Self-Rated Health				Depressive Symptoms			
	*n*	Adjusted RR (95% CI)		*p*-Value	*n*	Adjusted RR (95% CI)		*p*-Value	*n*	Adjusted RR (95% CI)		*p*-Value
Have contact with a lot of neighbors (≥20)	268	1.00			238	1.00			142	1.00		
Have contact with quite a lot of neighbors (10–19)	239	1.73	(0.84–3.55)	0.136	212	1.02	(0.71–1.46)	0.932	95	1.13	(0.58–2.20)	0.713
Have contact with some neighbors (5–9)	301	1.80	(0.81–4.04)	0.151	279	1.00	(0.69–1.44)	0.993	150	1.37	(0.80–2.35)	0.246
Have contact with just a few neighbors (<5)	118	2.33	(1.11–4.85)	0.025	116	1.27	(0.73–2.20)	0.402	44	1.37	(0.74–2.55)	0.313
Don’t know my neighbors	3	5.10	(1.02–25.6)	0.048	2	3.69	(1.06–12.8)	0.040	1	not applicable *		

Social capital at the community and individual levels, age, sex, marital status, cohabitants, educational attainment, history of major diseases (having any one of the following diseases: Stroke, myocardial infarction /angina, diabetes, vertebral or femoral neck fracture, Parkinson’s disease, or cancer), current drinking and current smoking were included to the model (Model 3). Note: RR = relative risk; CI = confidence interval; * not applicable because of the insufficient number.

## Data Availability

The datasets generated and analyzed during the current study are not publicly available because we did not receive consent for data provision to the third party.

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
