# Peer review of "Association of Structural Social Capital and Self-Reported Well-Being among Japanese Community-Dwelling Adults: A Longitudinal Study"

_ijerph, 2021, doi:10.3390/ijerph18168284_

Round 1

Reviewer 1 Report

  1. This research aims to examine whether structural social capital in the local community 14 was related to the future well-being of Japanese people aged 40 or above.
  2. Firstly, I think the contribution of the research should be more explicitly stated.
  3. Please, indicate in the introduction the objectives of your research since they are not clearly indicated.
  4. Moreover, I consider that the interest of the subject is not introduced well enough.
  5. The theoretical background that supports your objectives is not convincing enough. Please add more arguments to this section.
  6. I consider that the sample should be better characterized. In line with this, why did you decide to include only people aged 40 or above.
  7. Why did you decide to use a single-item to measure happiness?
  8. What method did you use to translate your measures?
  9. Why did you only analyze individuals aged 65 or above for depressive symptoms?
  10. The discussion of the results should be more developed.
  11. What are the theoretical implications of the research?
  12. The implications for practice should be more developed and clearly stated.
  13. You decided to not formulate any hypotheses. Could you please explain the reason why?
  14. I suggest you take another pass through the manuscript to clean up minor grammar and usage issues. Otherwise, the manuscript reads well.

Author Response

Dear Reviewer 1

Thank you for reading our manuscript carefully and giving useful comments. 

Please see the attachment for our revision.

Reviewer 2 Report

This is a very interesting paper. Revision is required for the following points.

First, please add previous studies on the relationship between social capital and subjective wellbeing.

Second, at the individual level, It is necessary to explain whether it is reasonable to set five persons as the criteria for judging the large and small of neighbors in contact to 5 people. The number of small group members is too small. This affects the analysis results.

Third, various variables are presented in table1. A regression analysis should be added for comparison between Time1 and Time2 while controlling for “demographic” variables and social capital as the independent variables and setting subjective wellbeing as the dependent variable.

Author Response

Dear Reviewer 2

Thank you for reading our manuscript carefully and giving useful comments. 

Please see the attachment for our revision.

Round 2

Reviewer 1 Report

I believe the manuscript has been significantly improved and now warrants
publication in IJERPH.

Author Response

Dear Reviewer 1,

We appreciate you checking our revised manuscript. Thanks to all reviewers and editors, we believe that the manuscript has become almost ready. It is all because of your kind help and advice.

Sincerely yours, Kazuya Nogi.